# Differential Expression of RSK4 Transcript Isoforms in Cancer and Its Clinical Relevance

**DOI:** 10.3390/ijms232314569

**Published:** 2022-11-23

**Authors:** Sisi Chen, Michael J. Seckl, Marc P. G. Lorentzen, Olivier E. Pardo

**Affiliations:** Division of Cancer, Department of Surgery & Cancer, Hammersmith Campus, Imperial College London, London W12 0NN, UK

**Keywords:** RSK4, *RPS6KA6*, isoforms, normal tissue, cancer, prognosis

## Abstract

While we previously revealed RSK4 as a therapeutic target in lung and bladder cancers, the wider role of this kinase in other cancers remains controversial. Indeed, other reports instead proposed RSK4 as a tumour suppressor in colorectal and gastric cancers and are contradictory in breast malignancies. One explanation for these discrepancies may be the expression of different RSK4 isoforms across cancers. Four RNAs are produced from the RSK4 gene, with two being protein-coding. Here, we analysed the expression of the latter across 30 normal and 33 cancer tissue types from the combined GTEx/TCGA dataset and correlated it with clinical features. This revealed the expression of RSK4 isoforms 1 and 2 to be independent prognostic factors for patient survival, pathological stage, cancer metastasis, recurrence, and immune infiltration in brain, stomach, cervical, and kidney cancers. However, we found that upregulation of either isoform can equally be associated with good or bad prognosis depending on the cancer type, and changes in the expression ratio of isoforms fail to predict clinical outcome. Hence, differential isoform expression alone cannot explain the contradictory roles of RSK4 in cancers, and further research is needed to highlight the underlying mechanisms for the context-dependent function of this kinase.

## 1. Introduction

The p90 ribosomal S6 protein kinases (RSKs) form a family of highly conserved Serine/Threonine kinases that function downstream of the mitogen-activated protein kinase (MAPK) signalling pathway [1]. The MAPK pathway is commonly hyperactivated in malignant cells, which contributes to their survival, cancer progression, and treatment resistance [2]. In addition, RSKs themselves display a variety of biological functions with relevance to the progression of cancer [1]. Therefore, they are generally perceived as attractive therapeutic targets [3]. Four RSKs exist in humans, RSK1-4, which share a high degree of sequence similarity (73–80%) [1,3,4,5]. All RSKs contain two evolutionarily conserved kinase domains, the C-terminal and N-terminal kinase domains (CTKD and NTKD, respectively), which show very high conservation [3,6,7]. Although this could have suggested functional redundancy, the reported roles of these kinases in cancer widely differ, with RSK1 and RSK2 being considered tumour promoters, while RSK3 is generally thought as a tumour suppressor, and the role of RSK4 remains a matter for debate [8,9,10,11,12,13].

RSK4 differs from other RSKs in a few key aspects. Firstly, RSK4 has substantially lower expression levels than RSK1-3 and is primarily expressed during embryonic development [1,14]. Secondly, unlike other RSKs, which remain inactive in the cytoplasm in the absence of stimuli and translocate to the nucleus upon activation [15,16,17], RSK4 is reported to be constitutively active in the cytoplasm of most cell types [18]. These differences in localisation and basal activity suggest that RSK4 may carry distinct functions from other RSKs. However, the biological functions of RSK4 have yet to be systematically elucidated.

In particular, the role of RSK4 in cancer remains controversial [5,9,11]. Indeed, this kinase was found to be a tumour promoter in some cancers, with increased expression levels correlating with poor prognosis, drug resistance, and metastasis in oesophagus, lung, bladder, kidney, and brain cancers [5,19,20,21,22]. In contrast, RSK4 was identified as a tumour suppressor in other malignancies, such as colorectal and gastric cancers, as well as acute myeloid leukaemia (AML), where its overexpression induced cell cycle arrest, inhibited cell invasion and metastasis, and was associated with better patient survival [23,24,25,26,27]. Finally, contradictory findings were reported for the role of RSK4 in the same cancer types (e.g., breast cancer), suggesting that its function in cancer progression is context-dependent [12,28,29,30,31,32,33].

One possible explanation for these inconsistencies could be the differential expression of its transcript/protein variants in different tissues [20]. The human RSK4 gene (*RPS6KA6*) produces at least four mRNA transcripts, of which two are protein-coding. The two protein-coding transcripts, *RPS6KA6-201* and *RPS6KA6-204*, encode RSK4 isoforms 1 and 2, respectively, which are generated by the use of an alternate first exon (exons 1B and X in isoform 1 and exon 1A in isoform 2). Both protein isoforms, however, have the same length (745 amino acids) and only differ in the sequence of their first ~100 amino acids. This difference might have functional relevance as in silico analysis suggests that RSK4 isoform 1 contains a myristylation site at its N-terminus, which is absent in isoform 2 and may bestow different subcellular localisation to these proteins [34]. Therefore, we hypothesised that the conflicting role of RSK4 in cancers may be explained by the differential expression of its isoforms.

Our study aimed to examine the expression pattern of the two protein-coding isoforms of RSK4 between healthy and tumour samples. To accomplish this, we used a combined RNA-Seq dataset from The Cancer Genome Atlas (TCGA) [35] and the Genotype-Tissue Expression project (GTEx) [36] to assess the expression of RSK4 isoforms 1 and 2 in 33 cancer types and their matching 30 healthy tissues. We also correlated isoform expression with clinical features, such as overall survival, pathologic stage, and metastatic and immune infiltration status, to reveal potential association with tumour progression.

## 2. Results

### 2.1. Expression Pattern of RSK4 Transcripts in Normal and Cancer Tissues

Based on the RNA-Seq data of the TCGA-GTEx combined cohorts, we found that the total RSK4 expression varied widely across different tissues and cancer samples (Figure 1A). RSK4 expression was highest in the pituitary gland, at almost three times the average expression for all tissues, whereas it was lowest in blood, where it is almost undetectable. Among cancer samples, RSK4 expression was highest in tumours originating from the adrenal gland (PCPG), followed by prostate adenocarcinoma (PRAD) and kidney chromophobe (KICH), while lowest in acute myeloid leukaemia (LAML). We then examined the expression of the four reported transcripts from the *RPS6KA6* gene. This revealed that the two protein-coding transcripts (isoforms 1 and 2) were predominantly expressed across both normal and malignant samples (Figure 1A), with isoform 1 always the most abundant (≥50% of the total transcript expression) (Figure 1A).

To investigate how the expression of RSK4 isoforms 1 and 2 varies within different normal tissues and cancer types, we analysed the TCGA and GTEx datasets separately. Kruskal–Wallis testing revealed that the expression of isoforms 1 and 2 varies significantly among both normal tissues and cancer types (*p* < 2.2 × 10^−16^ and *p* < 2.2 × 10^−16^, respectively) (Appendix A). Using the 25% and 75% expression quantiles, we stratified normal tissues and cancer types into low, medium, and high expressors (Figure 1B). Among normal tissues, we found that the ovary, thyroid, and blood vessels expressed high levels of both RSK4 isoforms, whereas the liver expressed low levels of both isoform 1 and 2. Interestingly, KICH was the only cancer type where both isoforms were highly expressed (Figure 1B). In contrast, Uterine Corpus Endometrial Carcinoma (UCEC), Head and Neck Squamous Cell Carcinoma (HNSC), Diffuse Large B-Cell Lymphoma (DLBC), Skin Cutaneous Melanoma (SKCM), Cervical squamous cell carcinoma and endocervical adenocarcinoma (CESC), Liver Hepatocellular Carcinoma (LIHC), and LAML showed low expression of both isoforms. In addition, we found that the spread of the expression for the isoforms was wider in cancer tissues compared to their normal counterparts, with wider interquartile ranges, suggesting unconcerted dysregulation of these transcripts during tumorigenesis. Furthermore, the spread of isoform 2 expression was invariably larger than that of isoform 1. Taken together, these results revealed that both RSK4 isoforms 1 and 2 show tissue and cancer-type-specific expression patterns, with malignant tissues demonstrating more widespread perturbations.

### 2.2. Comparison of RSK4 Isoform Expression between Tumour Samples and Their Matched Normal Counterparts

In general, comparing isoform 1 and 2 expressions between normal tissues and cancer samples revealed that isoform 1 was significantly downregulated in tumour samples (TCGA) as compared to normal tissues (GTEx) (*p*val < 2.2 × 10^−16^ Wilcoxon test), while no significant difference was observed for isoform 2 expression (Figure 2A). To further investigate changes in isoform expression associated with cancer development, we matched each cancer type with its normal counterparts based on the primary sites of tumours of origin (Appendix A).

Among the 19 normal/cancer pairs, the expression of both isoforms (1 and 2) was often either significantly upregulated or downregulated in cancerous tissues (*p.adj* < 0.001) with no apparent consensus pattern (Figure 2B). Nearly 90% (17/19) and 74% (14/19) of the pairs showed dysregulated isoform 1 or isoform 2 expression, respectively. We applied Cohen’s test to estimate the effect size of differences and found that some cancer types showed significantly upregulated expression of both isoforms with *p.adj* < 0.001 and effsize > 0.5, such as skin, cervical, and uterus cancers (Figure 2C). Moreover, some cancer types showed increased isoform 1 expression in tumours, with a corresponding decrease in expression for isoform 2, such as brain cancers. Others, however, have upregulated isoform 2, while isoform 1 expression is downregulated, such as Adrenal gland tumours. Hence, there is no obvious consensus that a change in expression for one particular isoform or their ratio is associated with cancer development. Moreover, the pattern of changes did not explain the reported roles of RSK4 as a tumour suppressor or promoter. Indeed, oesophagus, lung, bladder, kidney, and brain cancers, where RSK4 has been reported as a tumour promoter [5,19,20,21,22], showed very different patterns of isoform expression changes between malignant and corresponding normal tissues (Figure 2B). Hence, a change in the expression of either isoform or of their ratio did not explain the perceived function of RSK4 in particular cancers. Instead, the observed patterns may be indicative of the particular biology of these cancer types or their clinical characteristics, and this requires further investigation.

### 2.3. Association between RSK4 Isoform and Patients’ Survival

As expression patterns per se failed to inform on cancer types where RSK4 is considered a tumour promoter or suppressor, we next assessed whether this distinction could be explained by the association of isoform expression with clinical features of the disease. First, we applied the Cox proportional hazards model to assess the possible association between isoform expression and patient survival. We carried out both univariate and multivariate analyses (Figure 3A and Appendix A), using *p.adj* < 0.05 as a cut-off. According to univariate analysis, high isoform 1 expression is associated with better survival in patients with Brain Lower Grade Glioma (LGG), KICH, Adrenocortical Cancer (ACC), and Kidney Clear Cell Carcinoma (KIRC), but is associated with worse survival in patients with Stomach Adenocarcinoma (STAD) (Figure 3A). Conversely, high isoform 2 expression is associated with better survival in KIRC and Rectum Adenocarcinoma (READ) but worse survival in STAD and CESC. No significant correlation was found between patients’ survival and the ratio of isoform 1 and 2 expression. The fact that high levels of both isoforms 1 and 2 can be indicators of good prognosis in KIRC patients but poor prognosis in STAD patients suggests that RSK4 isoforms function in a cancer-type-specific manner. Interestingly, data from our survival analyses are in conflict with the reported effect of RSK4 as a tumour promoter/suppressor in those two tumour types [23,24,25,26,27].

We then performed multivariate analysis, which took into account covariates potentially influencing patient survival, to further evaluate the potential of isoforms 1 and 2 as independent prognostic factors. Those covariates included age (>60 or <60), gender, family history of cancer, pathological stages, metastasis, and cancer recurrence (Appendix A). Only four cancer types still exhibited significant associations between RSK4 isoform expression and survival following multivariate analysis (Appendix A). High isoform 1 expression was associated with longer survival of LGG (HR = 0.77) but worse survival of STAD patients (HR = 1.08), and high isoform 2 expression was associated with better survival of KIRC (HR = 0.93) but worse survival of CESC patients (HR = 1.23) (Figure 3B).

### 2.4. Correlation with Clinical Features

Having identified four cancer types where patients’ survival was significantly associated with RSK4 isoform expression (LGG, STAD, KIRC, and CESC), we speculated that this may reflect an association with particular clinical features. Cancer recurrence was linked to a slight but statistically significant decrease (*p* < 0.05) in isoform 1 expression in LGG patients (Figure 4A,B). Moreover, we found that reduced isoform 1 expression was associated with advanced pathologic stages (stage III and IV), metastasis, and cancer recurrence in KIRC patients (Figure 4A,B). These observations may partially explain why the high expression of isoforms 1 or 2 was linked with better survival in LGG and KIRC patients. However, no significant association between RSK4 isoform expression and clinical features was identified in STAD and CESC patients, suggesting the existence of additional yet undiscovered factors that act downstream of RSK4 isoform expression to impact patient survival.

Recently, the role of infiltrating immune cells on clinical outcomes of cancer patients, especially in the context of immune checkpoint therapy, has been an intense focus of research [37]. However, the potential role of RSK4 in regulating immune cell infiltration has not yet been investigated. Here, we used deconvolution algorithms on bulk RNA-Seq data for LGG, KIRC, STAD, and CESC to assess if changes in patient outcome could be attributed to the association between RSK4 expression and particular infiltrating cell types. We tested and compared five existing deconvolution algorithms, including CIBERSORT [38], xCell [39], EPIC [40], quanTIseq [41], and MCPcounter [42]. However, a recent study comparing predictions with ground truth suggested that the EPIC algorithm was the most reliable in its performance [43]. Therefore, we primarily focused on the results from this algorithm to correlate the expression of RSK4 isoforms with tumour-infiltrating cell types.

Examples of the output from the EPIC algorithm showing the relative abundance of B, natural killer (NK), CD4^+^ T and CD8^+^ T cells, macrophages, and cancer-associated fibroblasts (CAFs) in a selection of tumour samples are shown (Figure 4C). This analysis revealed that a higher proportion of CD4^+^ T cells is linked to increased survival in LGG, KIRC, and CESC patients (Figure 4D and Appendix A). In contrast, among the LGG, STAD, and KIRC patients, a larger proportion of CAFs was associated with worse survival (Figure 4D and Appendix A). The longer survival of LGG patients with high levels of CD4^+^ T cells (*p* < 0.0001) was associated with a positive correlation between the fraction of CD4^+^ T cells and isoform 1 expression (R = 0.34, *p* = 2.1 × 10^−15^) (Figure 4E). In contrast, the poorer survival of STAD patients with a larger proportion of CAFs (*p* = 0.039) was associated with a positive correlation between isoform 1 expression and CAF fraction (R = 0.14, *p* = 0.0042) (Figure 4D,E). These results suggested that the increased infiltration of CD4^+^ T cells or CAFs may contribute to better or worse survival of LGG and STAD patients with high RSK4 isoform 1 expression, respectively. This possibility was further supported by pathway enrichment analysis for genes co-expressed with RSK4 isoforms in these two tumour types (See Appendix A). We found that co-expressed genes in LGG are significantly associated with modulation of T cell activation (*p.adj* < 0.05) (Figure 4F), whereas those in STAD are enriched in modulators of CAF activation and biological functions [44,45,46] (Figure 4G).

Taken together, our results suggest that expression levels for RSK4 isoforms 1 or 2 alone fail to explain the tumour promoter/suppressor role of this kinase in different tumour types or associated clinical outcomes. However, isoform expression correlated with various clinical characteristics, including pathological stage, metastasis, cancer recurrence, and cancer infiltration by CD4^+^ T cells and CAFs in a limited number of tumour types where they predicted patients’ survival. Additional research is therefore needed to elucidate what determines the cancer-type-specific functions of RSK4.

## 3. Discussion

As one of the downstream mediators of MAPK signalling, RSK4 has been found to be involved in regulating progression, metastasis, and drug resistance in multiple malignancies [3,5,11,20]. However, whether RSK4 is a tumour promoter or suppressor remains controversial, with conflicting reports in different cancer types [5,9,11]. Since RSK4 can be encoded by two transcript isoforms with unknown differences in function, we hypothesised that reported discrepancies for the role of RSK4 in cancer may be influenced by differential expression of the RSK4 isoforms.

Here, we undertook the first in-depth analysis of the expression patterns of RSK4 mRNA isoforms among normal and tumour tissues, using the RNA-Seq datasets from GTEx and TCGA. We found that the two protein-coding isoforms of RSK4, isoforms 1 and 2, were the predominantly expressed of the four RSK4 transcripts, with their expression varying significantly across normal and tumour samples. For instance, both isoforms displayed elevated expression in cancers originating from the breast, lung, muscle, pancreas, and prostate, while they were downregulated in cancers of the oesophagus, skin, testis, thyroid, and uterus and unchanged in the remaining cancers. Hence, increased expression of RSK4 mRNA is not a general feature associated with the transformation process. This finding may be taken to suggest that the therapeutic usefulness of RSK4 targeting would be limited to only a subset of cancers which overexpress this kinase. However, we have previously demonstrated that selectively inhibiting RSK4 shows therapeutic benefits in in vivo and ex vivo models of lung and bladder cancers, respectively [20]. While we show here that lung cancer has increased mRNA expression of both RSK4 isoforms 1 and 2, bladder cancer has not. Hence, overexpression of RSK4 is not a pre-requisite for the therapeutic effectiveness of inhibiting this kinase.

Some normal tissues showed high expression levels of both isoforms (i.e., pituitary, thyroid gland, blood vessels, and ovary), while others showed lower expression than average (i.e., liver and spleen). This may suggest more prominent physiological roles for RSK4 in particular tissues, and a tissue-promoter-specific RSK4 knockout animal model would be useful to assess this possibility further. Furthermore, the fact that some tissues, such as the brain, show a high level of one isoform but low expression of the other may suggest differential functions of the two isoforms in normal biology. The systematic assessment of specific substrates and interactors for both isoforms, using analogue-sensitive kinase mutants [47] and tandem-affinity purification [48], respectively, would reveal these differences.

Following survival analysis, we identified statistically significant associations between RSK4 isoform expression and overall survival in LGG, KIRC, STAD, and CESC patients (*p.adj* < 0.05), suggesting that RSK4 isoforms are independent prognostic markers in these malignancies. However, the impact of each isoform on survival appears opposite between cancer types. Indeed, while higher expression of isoform 1 was associated with improved survival in LGG patients, it predicted worse survival in STAD patients (Figure 3B). Similarly, high isoform 2 expression was associated with better prognosis in KIRC patients but a worse prognosis in CESC patients (Figure 3B). These findings disprove our initial hypothesis that the cancer type-specific function of RSK4 could be explained by which isoform of this kinase is dominantly expressed. Instead, they reveal that the role of RSK4 isoforms themselves varies across cancer types. However, this may reflect the context-dependence of RSK4 signalling downstream of distinct oncogenic drivers or locally available growth factors. Future analysis should focus on elucidating the oncogenic pathways upstream of RSK4 that are shared between cancer types where it acts as a tumour promoter or suppressor to better understand what determines the differential outputs from this kinase.

There are several discrepancies between our results and the published literature. For instance, Li et al. observed RSK4 to be upregulated at both mRNA and protein levels in tumour biopsies from oesophagus cancer patients, and this upregulation was associated with radio resistance and poor prognosis [23]. In contrast, the results from our study showed that mRNA expression for both RSK4 isoforms was significantly downregulated in oesophagus cancer samples compared to their normal tissue counterparts and showed no association with patients’ survival. Similarly, while we ourselves published that increased expression of RSK4 in lung adenocarcinoma was associated with worse patients’ overall survival [20], we do not find this to be the case for individual isoforms in this study. Likewise, while high RSK4 expression was associated with poor prognosis in LGG and KIRC patients [25,26,27], the opposite was found here. Moreover, RSK4 was reported to be a tumour suppressor in gastric cancer (STAD) [24], but our study found that high RSK4 isoform 1 expression correlates with worse survival in STAD patients. Finally, while earlier studies reported both up- and downregulation of RSK4 expression in breast cancer, we found that mRNA levels for both RSK4 isoforms are significantly elevated in malignant breast tissues of all subtypes compared to normal tissue samples. These discrepancies may be linked to differences in methods used for measuring mRNA expression between studies or the origin and number of samples analysed as, for instance, our prior report was based on microarray data while the present study relies on RNA-sequencing from a different patient’s cohort. Furthermore, the choice of what is considered normal tissue may influence the conclusions reached as most studies used tumour-adjacent “normal” tissues as normal counterparts for comparison with cancer samples (for instance, see (Li et al., 2020) [23]). However, numerous reports show how the presence of a malignant tumour influences the expression patterns, behaviour, and signalling of adjacent normal tissue [49]. Hence, we believe that using GTEx expression data from truly normal tissues is more robust for such comparisons rather than the tumour-adjacent “normal” samples from the TCGA dataset.

In addition to revealing the relationship between RSK4 isoforms and clinical features, our study provided preliminary evidence supporting the potential involvement of RSK4 isoforms in regulating the tumour microenvironment. We used the EPIC algorithm to deconvolve the bulk RNA-Seq data from TCGA samples and predict the infiltrating cellular landscape in various cancer types. This suggested a positive correlation between RSK4 isoform expression and the relative proportion of B cells, CD4^+^ T cells, and CD8^+^ T cells (*p* < 0.05) among cancer types where high isoform 1 or 2 expression associates with better survival (LGG and KIRC). In contrast, in cancers where RSK4 expression associates with worse survival, a positive correlation between isoform expression and relative fraction of CAFs was found (*p* < 0.05) (Figure 4D). Pathway enrichment analysis for genes co-expressed with RSK4 in these cancers further supported these conclusions, as these associated with signalling are directly relevant to the predicted infiltrating cell types (Figure 4F,G). These findings suggest that RSK4 may help structure the tumour microenvironment, including infiltrating immune cells, a hypothesis that should be experimentally tested as this could explain the reported discrepancies for the role of this kinase in various tumour types.

However, our study suffers from several limitations. First, we only studied the mRNA expression for RSK4 isoforms, which may not accurately reflect the protein expression or activity of this kinase. Other factors, such as RNA editing and translation efficiency, protein post-translational modification and localisation, may all impact the functions of RSK4 isoforms in cancer. Future studies should therefore focus on RSK4 isoform protein expression and localisation in different cancer types, which will require developing isoform-specific antibodies. An additional challenge is the fact that a single RSK4 cDNA was found to be able to generate several protein variants with molecular weights ranging from 33 to 135 kDa [50]. In addition to focusing on mRNA expression, we did not study the association of the two non-protein coding mRNA isoforms of RSK4 with clinical features. Doing so may be relevant as non-protein-coding mRNAs have been found to regulate several processes, including transcription and translation of protein-coding mRNAs [51]. Finally, while the EPIC algorithm was reported to be the most robust of available bulk RNA-Seq deconvolution methods [43], its predictive accuracy may vary across cancer types due to tumour-selective bulk mRNA expression profile effects.

## 4. Material and Methods

### 4.1. Data Sources

The transcript expression RNAseq data in the GTEx (https://gtexportal.org/home/) and the TCGA (https://portal.gdc.cancer.gov/) were retrieved from the TCGA TARGET GTEx study of UCSC Xena Toil RNA-Seq Recompute Compendium (https://xenabrowser.net/) (accessed on 31 January 2022) [52]. The transcript expression RSEM TPM dataset was downloaded from UCSC Xena, containing 9807 samples from TCGA (with 33 different cancer types) and 7414 samples from GTEx (with 30 different normal tissues). The phenotype and survival data were also downloaded from UCSC Xena.

### 4.2. Statistical Tests

Since the isoform expression dataset is non-normally distributed, several non-parametric statistical tests, including Wilcoxon, Kruskal–Wallis and Dunn’s post-hoc tests, were used to conduct pairwise and non-pairwise comparisons between two or more groups of data. In addition, Cohen’s test was used to calculate the effect size as a measure of the standardised difference between two groups regardless of the sample size effect.

### 4.3. Survival Analysis

Cox proportional hazards regression model was used to assess the correlation between RSK4 isoform expression and overall survival (OS) of patients with different cancer types. Both univariate and multivariate analyses were performed to evaluate the potential of RSK4 isoform expression as an independent prognostic marker for specific cancer types. For multiple testing corrections, the *p*-value was adjusted using Bonferroni correction into False Discovery Rate (FDR), with an FDR ≤ 0.05 cut-off chosen for significance. To visualise the association between isoform expression and patients’ survival, Kaplan–Meier curves were plotted with patients stratified into high or low expressors using the median expression value as cut-off.

### 4.4. Cell Type Deconvolution Analysis

To estimate the abundance and type of infiltrating immune cells based on bulk-RNA-Seq data, we used the Immunodeconv R package, which deconvolves mRNA expression data using several competing algorithms, including CIBERSORT [38], xCell [39], EPIC [40], quanTIseq [41], and MCP-counter [42]. The input data used was a TPM-normalized, non-log-transformed expression matrix, with HUGO gene symbols as row names and sample IDs as column names, according to the algorithms’ guidelines. Since the output of the five algorithms differs in the number and nature of cell types inferred as well as in the numerical format (i.e., cell type fraction, percentage, empirical number), the relative fractions for common cell types inferred were calculated in each instance to enable cross-comparison of the predicted cell abundance. The relative fractions of B cells, CD4^+^ T cells, CD8^+^ T cells, macrophages, cancer-associated fibroblasts (CAFs), and natural killer (NK) cells were extracted and correlated with patients’ survival and RSK4 isoform expression.

### 4.5. Gene Co-Expression and Pathway Enrichment Analysis

Spearman’s rank correlation was performed to identify genes co-expressed with RSK4 isoforms in cancers where they predicted patients’ survival. The top 100 co-expressed genes with Rho > 0.55 and adjusted *p*-value < 0.05 were extracted and subjected to pathway enrichment analysis using the Kyoto Encyclopedia of Genes and Genomes (KEGG), PANTHER version 17.0, and the NCATS BioPlanet databases.

### 4.6. Data and Code Availability

Data analysis was mainly performed within Rstudio version 4.1.2. All data used or generated in our study are available in the Appendix A. All code is accessible at https://github.com/SisiChen16/RSK4-isoforms.git.

### 4.7. Concluding Remarks

In short, we provide here the first extensive bioinformatic investigation of the expression patterns of RSK4 mRNA isoforms in both normal and cancer samples. Our findings show that RSK4 protein-coding isoforms are associated with patient survival, pathological stage, cancer recurrence, metastasis, and immunological infiltration in a subset of tumour types. However, there is a lack of consensus on the impact of RSK4 total or isoform-specific expression on these variables across impacted tumour types. Further wet- and dry-lab studies are therefore required to better understand additional variables that influence the function of this kinase in cancer progression.

## Figures and Tables

**Figure 1 ijms-23-14569-f001:**
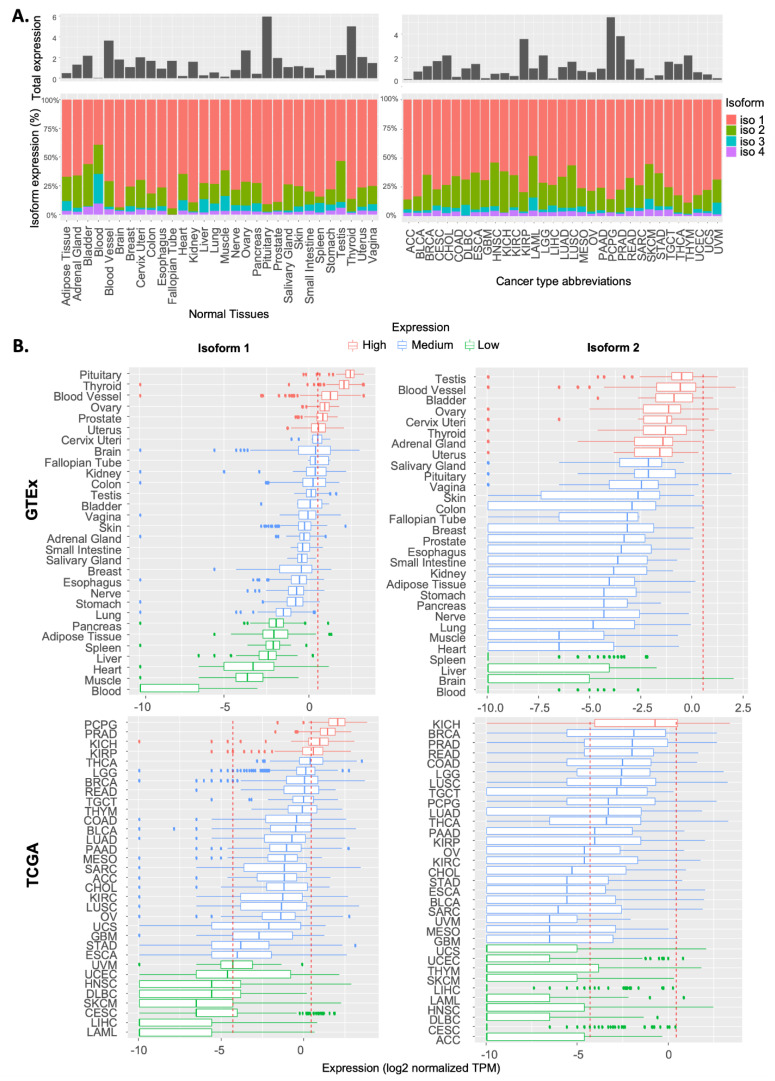
Expression of RSK4 transcripts in normal and cancer tissues. (**A**) Top: Histogram showing total RSK4 expression across normal (GTEx) (left) and cancerous (TCGA) (right) tissues. Bottom: Stacked bar plot illustrating the relative expression of the four transcript isoforms of RSK4. Tumour type abbreviations are decoded in Appendix A. (**B**) Boxplots showing the distribution of isoform 1 (left) and 2 (right) expressions among different normal tissues (top) and cancer types (bottom). Dotted lines mark the 25% and 75% quantiles used to categorise tissues into high (red), medium (blue), and low (green) expressors.

**Figure 2 ijms-23-14569-f002:**
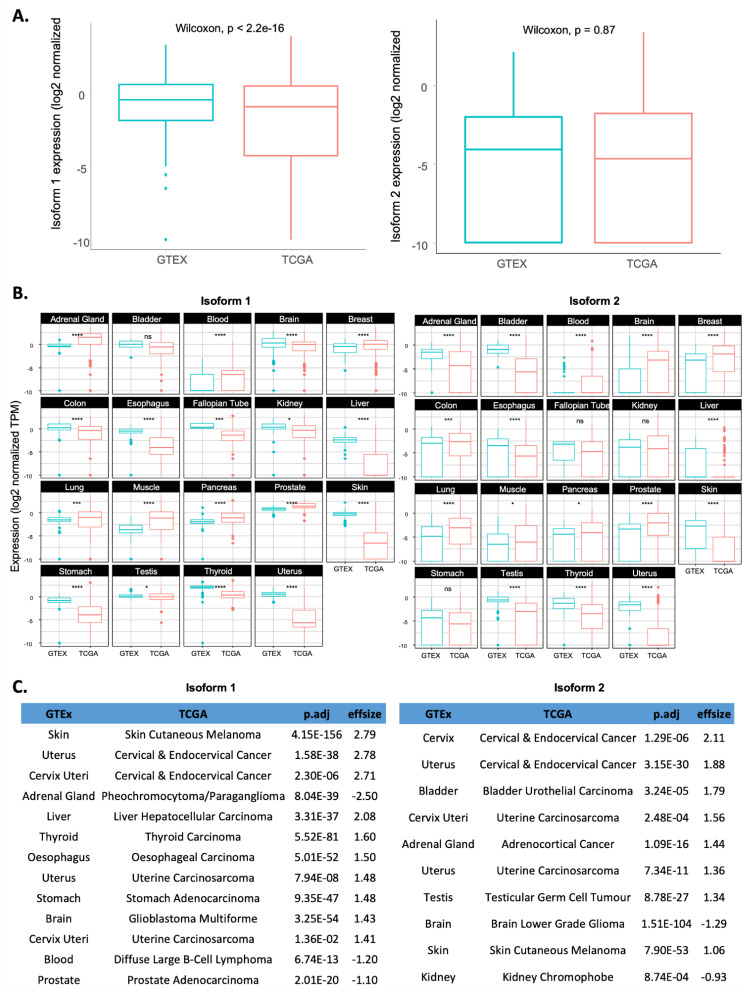
Comparison of RSK4 isoform expression between normal tissues (GTEx) and tumour samples (TCGA). (**A**) Comparison of overall isoform 1 and 2 expressions between normal and cancer samples. (**B**) Comparison of isoform 1 and 2 expressions between individual normal tissues and corresponding cancer types. Stats: Wilcoxon tests were used with * = *p.adj* < 0.05; *** = *p.adj* < 1 × 10^−3^; **** = *p.adj* < 1 × 10^−4^; *ns* = not significant. (**C**) Comparison of RSK4 isoform 1 (left) and 2 (right) expression between normal and corresponding cancerous tissues is shown for cases with large effect size (>0.5), as determined by Cohen’s test. Adjusted *p*-values are from Dunn’s test with effect size correction.

**Figure 3 ijms-23-14569-f003:**
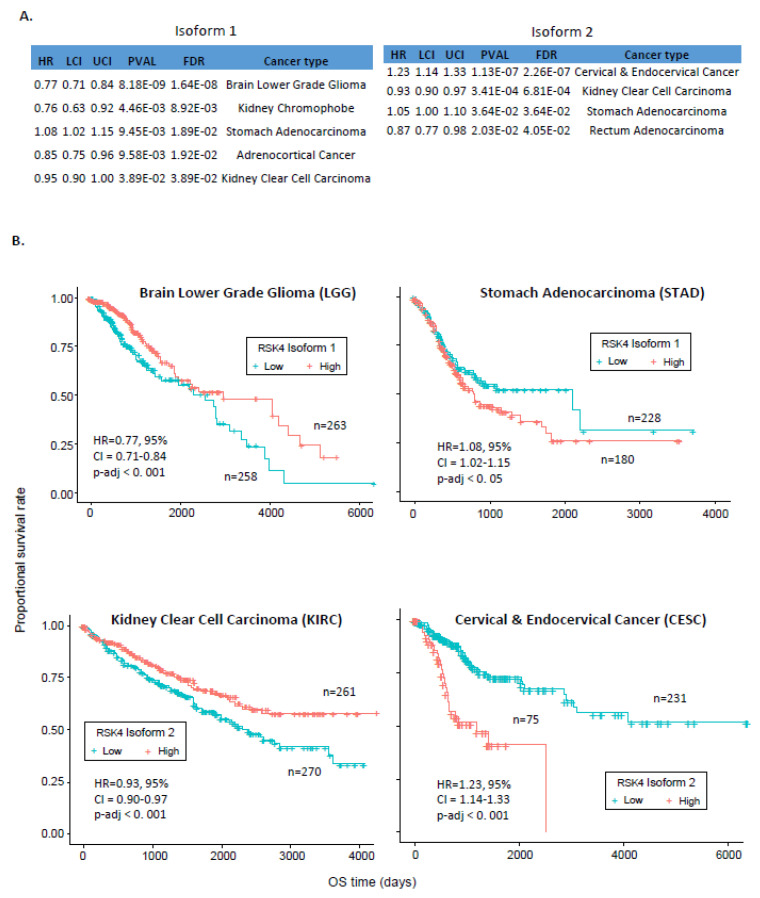
Expression of RSK4 isoform 1 or 2 correlates with overall survival in some cancer types. (**A**) Results from univariate analysis correlating the expression of isoforms 1 (left) and 2 (right) with overall survival across 33 cancer types. Only cases with significant correlation (FDR ≤ 0.05) are shown. Estimates of the Hazard Ratio (HR) and the Upper (UCI) and Lower (LCI) Confidence Intervals are shown. (**B**) Kaplan–Meier survival curves showing the correlation of high (red) and low (blue) expression of isoform 1 or 2 with overall survival in LGG, STAD, KIRC, and CESC patients. Median expression was used as cut-off.

**Figure 4 ijms-23-14569-f004:**
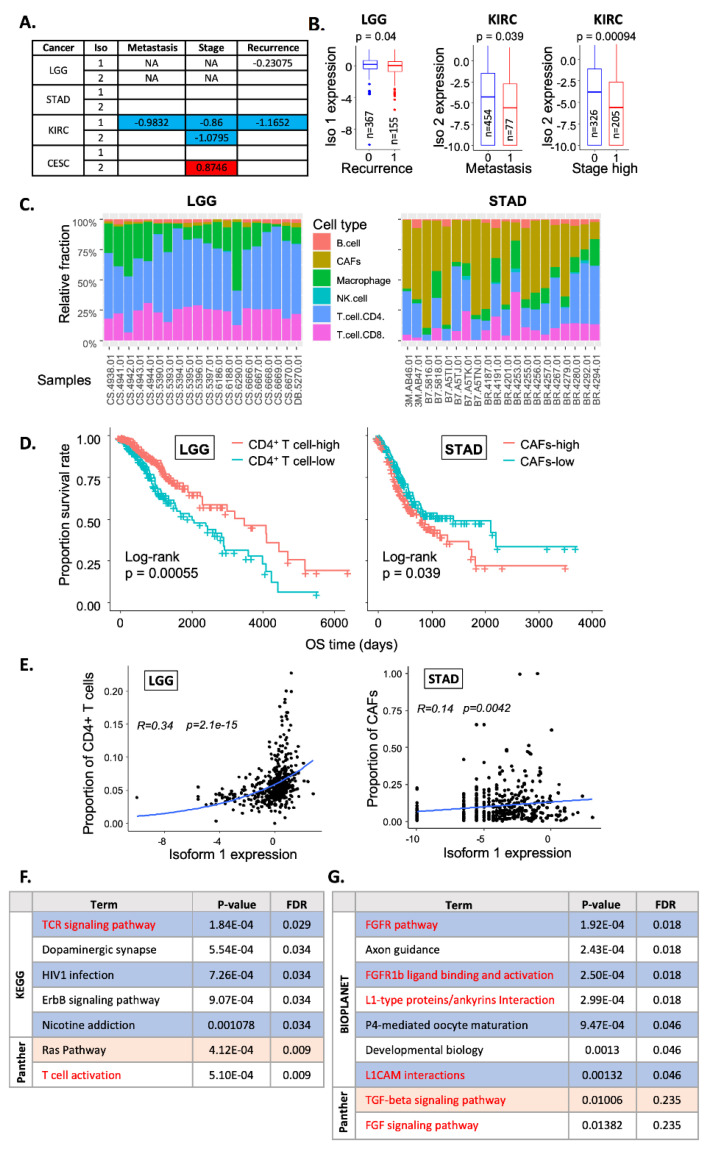
Correlation of RSK4 isoform expression with clinical features. (**A**) Spearman correlation of RSK4 isoform expression with the metastatic status, stage, and recurrence in LGG, STAD, KIRK, and CESC patients. Numbers represent correlation coefficients, with blue indicating statistically significant negative and red positive correlations. NA, clinical features not provided. (**B**) Box plot showing RSK4 isoform expression versus cancer recurrence in LGG, and metastatic status and high stage (III and IV) in KIRK patients. 0, no event; 1, event; n, number of patients. Statistics: Wilcoxon test. (**C**) Stacked bar plots of inferred cell type fractions from the EPIC deconvolution algorithm for the first 20 samples for LGG and STAD in the TCGA dataset. (**D**) Kaplan–Meier curves with log-rank test for the association between CD4^+^ T cell or CAFs fractions and the overall survival of LGG and STAD patients, respectively. (**E**) Spearman correlation was used to examine the correlation between isoform 1 expression and proportion of CD4^+^ T cells in LGG patients and CAFs in STAD patients. R represents correlation coefficient. (**F**,**G**) The top 100 genes co-expressed with RSK4 isoform 1 (Spearman correlation, Rho > 0.55 and adjusted *p*-value < 0.05) were subjected to pathway enrichment analysis in EnrichR and terms from databases showing *p* ≤ 0.05 are displayed. Terms highlighted in red are associated with T cell (**F**) and CAF (**G**) activation.

## Data Availability

All code is accessible at https://github.com/SisiChen16/RSK4-isoforms.git.

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
