# Peer review of "Differential Expression of RSK4 Transcript Isoforms in Cancer and Its Clinical Relevance"

_ijms, 2022, doi:10.3390/ijms232314569_

Round 1

Reviewer 1 Report

Heterogeneity between individual patients are challenges for cancer patient clinic management. Sensitive predictive biomarker(s) is crucial for the success of precision medicine. This is a well written report on an interesting extensive bioinformatic investigation of RSK4 mRNA isoforms in both normal and cancer samples with a great use of publicly available TCGA-GTEx database for predictive biomarkers. Although the conclusion is inconclusive, the authors showed thorough investigation approaches and presented reliable data. However, there are some concerns about this manuscript:

1.      mRNA expression level may not be as translationally relevant as the protein expression. Results should be confirmed with at least a small group of fresh or FFPE tissue by IHC staining, especially for a few cancer types with significant results.

2.      In addition to demographics (clinical outcome) tested, there is another very important aspect of cancer progression that should be considered, treatment responses (lines 72 and 173).

3.      It seems the normal and cancer tissue data used were paired for the disease tissue type, but not paired within each patient (at least was not clear for this reviewer). This may impair statistical analysis power.

4.      Figure 1A, no description of what items listed in the Tables

5.      Figure 2 legend: line 148, (D) should be (C).

6.      Figure 4A, red box for CESC is not mentioned and explained

7.      Line 188-198, the description of the results is a bit confusing. Line 192, Fig. 4A should be Fig. 4B, and Line 194, Fig. 4B should be Fig. 4A and 4B

Author Response

Reviewer #1

Heterogeneity between individual patients are challenges for cancer patient clinic management. Sensitive predictive biomarker(s) is crucial for the success of precision medicine. This is a well written report on an interesting extensive bioinformatic investigation of RSK4 mRNA isoforms in both normal and cancer samples with a great use of publicly available TCGA-GTEx database for predictive biomarkers. Although the conclusion is inconclusive, the authors showed thorough investigation approaches and presented reliable data.

We thank the reviewer for their positive outlook on our manuscript.

However, there are some concerns about this manuscript:

  1. mRNA expression level may not be as translationally relevant as the protein expression. Results should be confirmed with at least a small group of fresh or FFPE tissue by IHC staining, especially for a few cancer types with significant results.

The reviewer is correct in stating that mRNA and protein expression levels do not necessarily align and that IHC results on FFPE samples would strengthen our data. However, we do not currently have antibodies that can discriminate between RSK4 isoform 1 and 2, and so results obtained would represent combined RSK4 isoform expression that, in our own analysis, does not have predictive value. Therefore, we cannot currently perform such analysis.

  1. In addition to demographics (clinical outcome) tested, there is another very important aspect of cancer progression that should be considered, treatment responses (lines 72 and 173).

The reviewer is correct that it would have been valuable to perform this analysis. However, we felt that the quality of the clinical outcome data was insufficient to allow reliable assessment. Indeed, the “primary_therapy_outcome_success” column within the clinical data associated with the TCGA dataset fails to specify the treatment modality considered (surgery, chemo/radiotherapy, targeted therapy, etc), which makes it difficult to assess reasons for potential association between RSK4 expression and outcome. In addition, primary treatment modalities vary considerably between tumour types and having no information on the modality for which outcome was measured makes comparing the role of RSK4 in different tumour types impossible. Finally, this column contains a very large number of missing values with, for instance, over half of KIRC patients lacking information on treatment response. Taken together, we considered that these limitations made meaningful analysis impossible considering the focus of our manuscript.

  1. It seems the normal and cancer tissue data used were paired for the disease tissue type, but not paired within each patient (at least was not clear for this reviewer). This may impair statistical analysis power.

GTEx and TCGA samples do not originate from the same patients and there is therefore no way to pair normal and tumour samples for each patient. The TCGA dataset only contains a limited number of “normal” samples which are matched with the corresponding tumour samples for the same patients. However, these are normal adjacent samples, and, as outlined in our manuscript, treating these samples as normal can be misleading as the presence of the tumour is known to impact the biology of surrounding normal tissue. This is precisely the reason for us using GTEx samples in our study rather than normal adjacent samples from the TCGA. Hence, we have no way currently to pair tumour samples with true corresponding normal controls for the same patients.

  1. Figure 1A, no description of what items listed in the Tables

We are sorry for this oversight and have now described all tumour types’ abbreviations as part of Supplementary Table 1 and included reference to this in the legend of Figure 1A.

  1. Figure 2 legend: line 148, (D) should be (C).

We thank the reviewer for noticing this error and have now modified the legend accordingly.

  1. Figure 4A, red box for CESC is not mentioned and explained

We apologise for our lack of precision in the figure legend. Blue colour indicated statistically significant negative and red colour statistically significant positive correlations. We have now modified the legend to reflect this.

  1. Line 188-198, the description of the results is a bit confusing. Line 192, Fig. 4A should be Fig. 4B, and Line 194, Fig. 4B should be Fig. 4A and 4B

We agree that our reference to the figure panels was confusing and have now corrected this in the text of the manuscript.

Reviewer 2 Report

In this manuscript the further role of RSK4 isoforms in carcinogenesis was investigated. Authors analyzed the expression of  RSK4 transcripts  for 30 normal and 33 cancer types using  combined GTEx/TCGA dataset.  They made the conclusion that differential isoform expression alone cannot explain  the contradictory roles of RSK4 in cancers/  This bioinformatic research provides a new perspective for the experimental study. I liked that authors applied  TCGA and GTEx datasets using the 25% and 75% expression quantiles and stratified tissues into low, medium, and high expression levels. Such approach allows revealing differences in expression level. I would recommend accepting this manuscript.

Author Response

In this manuscript the further role of RSK4 isoforms in carcinogenesis was investigated. Authors analyzed the expression of  RSK4 transcripts  for 30 normal and 33 cancer types using  combined GTEx/TCGA dataset.  They made the conclusion that differential isoform expression alone cannot explain  the contradictory roles of RSK4 in cancers/  This bioinformatic research provides a new perspective for the experimental study. I liked that authors applied  TCGA and GTEx datasets using the 25% and 75% expression quantiles and stratified tissues into low, medium, and high expression levels. Such approach allows revealing differences in expression level. I would recommend accepting this manuscript.

We thank the reviewer for their positive outlook on our manuscript.